# Modeling Retinal Ganglion Cell Dysfunction in Optic Neuropathies

**DOI:** 10.3390/cells10061398

**Published:** 2021-06-05

**Authors:** Vittorio Porciatti, Tsung-Han Chou

**Affiliations:** Bascom Palmer Eye Institute, University of Miami Miller School of Medicine, Miami, FL 33136, USA; tchou@med.miami.edu

**Keywords:** retinal ganglion cell function, pattern electroretinogram, glaucoma, optic neuropathy

## Abstract

As in glaucoma and other optic neuropathies cellular dysfunction often precedes cell death, the assessment of retinal ganglion cell (RGC) function represents a key outcome measure for neuroprotective strategies aimed at targeting distressed but still viable cells. RGC dysfunction can be assessed with the pattern electroretinogram (PERG), a sensitive measure of electrical activity of RGCs that is recorded non-invasively in human subjects and mouse models. Here, we offer a conceptual framework based on an intuitive state-transition model used for disease management in patients to identify progressive, potentially reversible stages of RGC dysfunction leading to cell death in mouse models of glaucoma and other optic neuropathies. We provide mathematical equations to describe state-transitions with a set of modifiable parameters that alter the time course and severity of state-transitions, which can be used for hypothesis testing and fitting experimental PERG data. PERG dynamics as a function of physiological stimuli are also used to differentiate phenotypic and altered RGC response dynamics, to assess susceptibility to stressors and to assess reversible dysfunction upon pharmacological treatment.

## 1. Introduction

Neuroprotection in glaucoma and other optic neuropathies is an area of intense investigation in preclinical models [1,2,3,4,5,6,7,8]. Neuroprotective strategies include a large variety of methods (dietary, exercise, environment, pharmacological, molecular, radiation, stem cells, gene therapy) that target specific aspects of neuronal biology or nonspecifically target conditions such as elevated IOP, inflammation, or immune system malfunction that may eventually cause neuronal damage. It is becoming increasing clear that, in glaucoma and other optic neuropathies, cellular and axonal dysfunction often precede cell death [9,10,11,12,13]. This helps in focusing neuroprotective strategies in a time window of opportunity at which distressed but still viable cells can be rescued from irreversible stages of neurodegeneration and perhaps restored to their normal function [14].

Thus, sensitive assessment of retinal ganglion cell (RGC) function in preclinical models and investigation of the effects of stressors and neuroprotectants can provide a powerful tool for the early detection of neuronal dysfunction, as well as the prediction of disease progression with or without treatment. Here, we offer a conceptual framework to model progressive RGC dysfunction in glaucoma and other optic neuropathies based on non-invasive, high-throughput electrophysiological methods suitable for neuroprotection studies.

## 2. To Be or Not to Be, This Is the Question

Progressive RGC dysfunction and death can be thought to undergo six stages that can be defined in lay terms (Table 1) from stage 0 (healthy) to stage 6 (effaced). The stages of cell damage are based on an intuitive state-transition model such as the Markov decision process used for disease management in patients [15] that also includes stages of dying and effaced RGCs that are detectable with retinal imaging and are relevant to neuroprotection. Each stage is potentially identifiable in vivo with non-invasive electrophysiology and/or non-invasive imaging, which probe different aspects of RGC population. For a first approximation, a reduced electrical signal represents the activity of surviving neurons, while OCT thinning represents lost neurons. The ideal time window of opportunity for early detection, prediction of progression, and effective neuroprotection appears to be at stages 1–2 (functional tipping point) and at stage 3 (very sick). RGC dysfunction at stages 1–3 is unlikely to be detected with imaging. Stage 4 is identifiable with real-time imaging of apoptosing RGCs [16,17], and stages 4–5 are identifiable with OCT imaging, such as the loss of inner retina thickness. At stages 4–6, electrophysiology is expected to be at floor level and is no longer useful to monitor disease progression. Note that RGCs at different stages may coexist in the same population. Non-invasive electrophysiology represents the integrated response of the cell population subtended by the visual stimulus.

### 2.1. Non-Invasive Assessment of RGC Function

Visual function can be assessed with several methods [18] that depend on the activity of both retinal and post retinal pathways such as visually evoked potentials (VEP), reflex-based optomotor response (OMR) and operant training. Here, we focus on the pattern electroretinogram (PERG), which assesses RGC function directly. While there are other RGC-sensitive ERG methods, the PERG is the best understood and most sensitive technique which specifically depends on the presence of functional RGCs [19]. The PERG amplitude is proportional to the number of RGCs at a given retinal location, and the spectrum of spatial frequencies at which a PERG response can be generated corresponds to the size to RGC receptive field centers [20,21,22]. The PERG spatial resolution for gratings (acuity) and contract threshold correspond to corresponding behavioral measurements [23]. Crucially, the PERG is rapidly abolished after optic nerve crush and is altered early in glaucoma, before histological RGC loss [19]. Importantly for neuroprotection studies, the PERG signal has a signal-to-noise ratio of about 1 Log unit, which allows meaningful assessment of changes over time and treatment over the entire disease scale. The assessment of PERG changes is simplified by recent developments of PERG technology that allow simultaneous assessment of responses with high signal-to-noise ratio from each eye in mice using a single subcutaneous needle in the snout [24]. This eliminates the necessity of corneal electrodes that may damage the cornea, deteriorate eye optics and alter eye pressure, and also facilitates experiments based on comparison between the responses of the two eyes. An example of the recording set up used for human and mouse PERG, and an example of a corresponding PERG waveforms are shown in Figure 1.

### 2.2. Physiological Significance of Altered PERG Signal

The PERG has been extensively used in preclinical and clinical studies of glaucoma and optic neuropathies to detect RGC dysfunction in cross-sectional and longitudinal studies. Losses of PERG signal have been shown to be reversible after IOP-lowering treatment [25,26]. While these are important applications, they do not provide information on the nature of PERG changes. For example, a reduced PERG signal may be the result of the missing contribution of lost RGCs and dying RGCs, reduced contribution of dysfunctional RGCs, and the normal contribution of healthy RGCs in unknown relative proportions. An observable example of this originates from in vivo, real-time imaging of human and rodent glaucomatous retinas labeled with the fluorescent biomarker annexin 5, which identifies a subpopulation of apoptosing RGCs coexisting with apparently normal RGCs [27]. Figure 2A summarizes a simple model of progressive glaucoma in which, between time zero and time 1, a proportion of RGCs (30%) becomes sick, function at 50% of normal capacity and survive for a limited amount of time. At time 2, sick RGCs start dying while 30% of the remaining healthy RGCs become sick, with the process repeating over time. These events will be reflected in progressive loss of function (PERG amplitude) and inner retina structure (OCT thinning), with the former expected to anticipate loss to a greater extent compared to the latter.

The proposed model is based on a simple two-parameter exponential decay function H(t) = H0*(1−*b*)^t where H(t) is the number of healthy RGCs at a given arbitrary time unit t, H0 is the number of healthy RGCs at disease onset (100 RGCs in the example) and *b* is the continuous decay rate (*b* = 0.3 in the example of Figure 2A) at which healthy RGCs become sick. Assuming that there is a time lag τ from sick to dead conditions (τ = 2 in the example of Figure 2A), then the number of dead RGCs at a given time t will be D(t) = H0 − [1 − (1 − *b*)^(t − τ)], t > 2; D(t) = 0, 0 ≤ t ≤ 2. The number of sick RGCs at a given time t will be S(t) = H0*[1 − (1 − *b*)^2]*(1 − b)^(t − 2), t > 2; S(t) = H0[1 − (1 − *b*)^t], 0 ≤ t ≤ 2. At any given time, the overall function F of surviving RGCs will be proportional to the number of healthy RGCs plus the number of sick RGCs multiplied by a dysfunction factor *d* (*d* = 0.5 in the example) F(t) = H(t) + S(t)**d*. The overall surviving RGC population P will be proportional to the number of healthy RGCs plus the number of sick RGCs P(t) = H(t) + S(t) (Figure 2B). Thus, modifiable parameters are the continuous decay rate *b* at which healthy RGCs become sick, the time lag τ between sick and dead RGC, and the dysfunction coefficient *d* of sick RGCs. Changes in these factors generate a family of functions that can be used for hypothesis testing and modeling experimental data. The experimental data in Figure 2C obtained in DBA/2J glaucoma [28] could be reasonably well modeled with parameters *b* = 0.3, τ = 6.5 months, and *d* = 0.5. In Figure 2B, the horizontal distance between the decay curves of function and structure provides an estimate of the lifespan of sick RGCs, which represents a time window of opportunity for treatment to prevent RGC death. The vertical distance between the decay curves of function and structure provides an estimate of RGC dysfunction that is not accounted for by cell death, which represents an opportunity for treatment to restore RGC function. Longitudinal clinical data in early glaucoma patients [29] also show PERG functional decay, anticipating retinal nerve fiber thickness decay by several years. Analogue models may be hypothesized for a variety of conditions impacting the susceptibly and lifespan of RGCs together with their ability to generate electrical signals under a protracted degenerative process. 

### 2.3. PERG Dynamics and RGC Functional Properties

To have a better insight into the significance of PERG changes in optic nerve disorders, it is necessary to isolate the activity of surviving and still functional RGCs. This can be done by investigating how the PERG response changes over a range of conditions. The resulting response dynamics reflects the ability of functional RGCs to detect changing conditions and regulate their activity accordingly. Additional insight into the activity of functional RGCs is provided by the PERG response latency, which reflects the contribution of healthy and sick RGCs but not the missing contribution of dying and dead RGCs. PERG latency is defined as the time that elapses between the onset of the visual stimulus and the peak of the response. PERG amplitude and latency dynamics can be investigated for a variety of physiological stimuli and positive/negative stressors, eventually providing a panel of biomarkers that will be helpful to identify the cause of dysfunction and formulate predictions on disease progression with or without treatments. A number of neuronal function properties are reflected in changes in PERG response dynamics (Table 2). These properties, not mutually exclusive, are widely used in different contexts with varying semantics, but are pragmatically defined here as they provide a framework to test hypotheses of different RGC functional conditions as well as biomarkers for neuroprotection studies.

### 2.4. PERG Dynamics—Physiological Approaches

Physiological approaches such as changes in stimulus intensity/duration or body posture may result in different PERG changes that reflect the ability of functional RGCs to regulate their activity. Figure 3 summarizes conceptual examples of homeostasis, gain control, and adaptation.

## 3. Homeostasis

Homeostasis (Figure 3A) characterizes the ability of RGCs to maintain a constant response over a wide range of physical and biological perturbations. Altered homeostasis perhaps represents the earliest stage of RGC dysfunction that is reflected in the PERG. The detection of altered homeostasis helps in formulating an early diagnosis and predicting progression and may represent the rationale for neuroprotective intervention. Altered RGC homeostasis is thought to occur in pre-glaucomatous DBA/2J mice and patients with suspected glaucoma, where a normal PERG may become reduced in susceptible eyes but not in control eyes upon physiological IOP elevation during head down tilt [30,31]. PERG susceptibility upon head-down tilt in those with suspected glaucoma predicts thinning of retinal fiber layer thickness after 5 years [32].

## 4. Gain Control

Gain control (Figure 3B) characterizes the ability of RGCs to regulate their activity with increasing stimulus strength. Typically, the relationship between PERG amplitude/latency and stimulus contrast is not linear, implying regulatory mechanisms in the RGCs and/or in the inner retinal circuitry impinging on them to contain RGC response in an optimal dynamic range. Altered PERG contrast gain control means that RGCs function in a different way, which may represent a biomarker of functional remodeling to improve survival rather than heralding cell death. Examples of altered PERG contrast gain control dynamics of amplitude and latency are provided by experiments in different mouse models in which neurotrophic support has been disturbed without causing cell death, including intravitreally injection of brain-derived neurotrophic factor (BDNF), anti-BDNF or lesion of the superior colliculus [33]. These results represent a proof of concept that PERG dynamics could be used as a tool for in vivo monitoring of RGC functional plasticity and for phenotypic screening (see Figure 5).

## 5. Adaptation

Adaptation (Figure 3C) characterizes the autoregulatory ability of RGCs to reduce their activity to a lower level in response to sustained visual stimuli that cause high energy demand. Visual stimuli that induce strong PERG adaptation are steady-state, high-contrast reversing patterns in humans [34] and transient, high-contrast reversing patterns with superimposed flickering light in mice [35] that are associated with increased metabolic demand and vasodilation. PERG adaptation is reduced in human aging, glaucoma, optic neuritis and non-artheritic-ischemic-neuropathy (NAION) [36,37] and is also reduced in DBA/2J mouse glaucoma [38]. PERG adaptation can be restored in DBA/2J glaucoma with sustained diet that supports mitochondrial function [38].

## 6. PERG Dynamics—Interventional Approaches

Interventional approaches such as treatment or exposure to stressful conditions may result in a variety of PERG changes that reflect temporary changes in RGC function. Figure 4 summarizes conceptual examples of enhancement, restoration, susceptibility and resilience that can be applied to both standard PERG responses and PERG response dynamics.

## 7. Enhancement

Enhancement (Figure 4A) characterizes nonspecific improvement of the PERG signal after treatment. Both the normal PERG amplitude of the control group and the reduced PERG amplitude of the group with manifest pathology improve, meaning that the treatment boosts function of all RGCs independently of pathology. For example, citicoline (cytidine-5′-diphosphocholine) has been shown to improve the PERG amplitude in glaucoma and NAION patients [39], but has been also shown to induce sustained improvement of PERG amplitude in control subjects [40]. Enhancement of PERG amplitude in control human subjects is also reported to occur after oral administration of levodopa [41].

## 8. Restoration

Restoration (Figure 4B) characterizes specific improvement of PERG amplitude after treatment in the manifest pathology group but not in the control group. For example, Ventura and Porciatti [26] treated both controls and glaucoma patients with IOP-lowering medications. Comparable reduction in IOP in both groups resulted in PERG amplitude improvement in the glaucoma group but not the control group.

## 9. Susceptibility

The intrinsic susceptibility to glaucoma and optic neuropathies due to complex trait inheritance and aging is well known [42,43,44,45]. Susceptibility to exogenous factors (Figure 4C) is less investigated and less understood, and it is used here to characterize reversible reduction in an otherwise normal PERG amplitude in response to a variety of stressors (physiological, physical, chemical, molecular, etc.). Inducible, reversible PERG reduction can be used as a provocative test to investigate factors that impair RGC defense mechanisms. Examples include temporary IOP elevation in glaucoma [30,46], while inducible, reversible PERG loss that targets the optic nerve is provided by retrobulbar lidocaine injections in mice, which temporarily block axon transport in the optic nerve [47]. Postconditioning after repeated lidocaine injections results in endogenous upregulation of trophic factors in the retina that have a neuroprotective effect in the DBA/2J mouse glaucoma [48]. Investigation of susceptibility to stressors may represent a very promising field, as susceptibility to specific stressors provides the rationale and a target for neuroprotection.

## 10. Acquired Resilience

Body tissues have the ability to autoregulate protective or repair mechanisms, including adaptive remodeling, in response to a large variety of everyday exogenous stressors, similarly to the immune system, which make the tissue resilient to future stress of the same nature [49]. Repair mechanisms include adaptive remodeling during the course of the neurodegenerative disease [50]. Acquired resilience is used here to characterize resilience gained after specific treatments that minimize susceptibility to provocative tests (Figure 4D). So far, a few provocative tests have been proposed, such as head-down posture [30,31] or water drinking [51], which cause IOP elevation and may result in PERG reduction in susceptible eyes. Combined assessment of susceptibility to novel provocative tests and acquired resilience to the same provocative test after treatment may represent a promising field of investigation, as it would provide a predictive index of neuroprotective efficacy of treatments based on baseline assessment.

## 11. Functional Phenotyping

The need for functional phenotyping increases in parallel with the ever-increasing number of genetically engineered rodent models. Susceptibility and resilience in glaucoma have a strong genetic component [52]. For example, BXD mice (derived from crossing glaucoma-resilient C57BL/6J mice with glaucoma-prone DBA/2J mice) allow investigation of genetic regulation of factors associated with glaucoma risk [9,53]. One way to determine the RGC functional phenotype of genotypes with different resilience/susceptibility is to investigate the PERG contrast dynamics (Figure 5). At low contrast (0.3), PERGs of pre-glaucomatous C57BL/6J and DBA/2J mice have similar amplitude and latency. With increasing contrast, the rates of change in amplitude and latency of the two strains substantially diverge, indicating different contrast gain and integration time [54]. This suggests different functional circuitry in the inner retina that may be related to the larger RGC population in DBA/2J compared to C57BL/6J [55]. Contrast gain and integration time also change in the same phenotype upon genetic and induced changes in neurotrophic support [33], suggesting altered functional circuitry subserving RGC response.

## 12. Age-Related Factors

Neuroprotection studies are typically performed over an extended time period. Intrinsic, age-dependent functional changes need to be accounted for to isolate the effect of specific stressors and treatments. For example, as shown in Figure 6A, in D2 mice IOP increases with increasing age while the PERG amplitude progressively decreases to a floor at about 11 months [28]. Statistical analysis reveals that while there is a strong inverse correlation between PERG amplitude and IOP (r^2^ = 0.43, *p* < 0.001), age independently contributes (*p* < 0.0001) to progressive PERG loss. This implies that there are specific age-related factors that could be the target of neuroprotective strategies, in addition to IOP lowering [56]. Williams et al. [10] identified mitochondrial abnormalities as major driver of neuronal dysfunction. B2 mice treated with either vitamin B3 rich diet or gene therapy driving Nmnat1, a key NAD+-producing enzyme, did not develop glaucoma despite IOP elevation [10]. Similar results have been obtained with a diet enriched with pyruvate, which provides bioenergetic support [57]. Age-related factors also alter eye structure independently of IOP [58,59] (Figure 6B–D). Between 2.5 and 6 months of age, both B6 and D2 mice grow in weight and eye length, although after 6 months eye elongation and anterior chamber depth are considerably larger in D2 in association with increased IOP. Cone et al., 2010 [60] induced IOP elevation in B6, D2, and CD1 mice by an average of 4.4 mmHg with intracameral injections of microbeads, resulting in different eye elongation and RGC death in the three strains after 6–12 weeks. Altogether, spontaneous and induced change in eye size may alter biomechanical properties of eye tissues which may also play a role in RGC and axon susceptibility independently of IOP.

## 13. Conclusions

The conceptual framework we have discussed is based on the assumptions that (1) in the progression of glaucoma and optic neuropathies, reversible RGC dysfunction precedes RGC death, (2) RGC dysfunction and its dynamics can be assessed non-invasively with PERG, (3) the susceptibility to positive/negative stressors is reflected in PERG changes of the opposite sign. Although these assumptions have not been firmly demonstrated, they are supported by a substantial body of literature including the examples shown above and the proposed mathematical model that fits the experimental data. We realize that the pathophysiology and temporal dynamics of glaucoma and optic neuropathies are more complex than those we have assumed to minimize the contributing factors to the model. Future studies will prove or disprove this model and introduce additional models. We also realize that the PERG is a global metric and cannot distinguish local changes that are frequent in early manifest glaucoma and can be detected with imaging. Altogether, the proposed conceptual framework emphasizes several different RGC physiological conditions that can be characterized with PERG and used as biomarkers for staging the disease, for establishing sensitive functional endpoints at early stages of disease before irreversible RGC death, and for assessing the effect of neuroprotective treatments. PERG dynamics can also be used for phenotyping genetically engineered models with possible alteration of RGC function.

## Figures and Tables

**Figure 1 cells-10-01398-f001:**
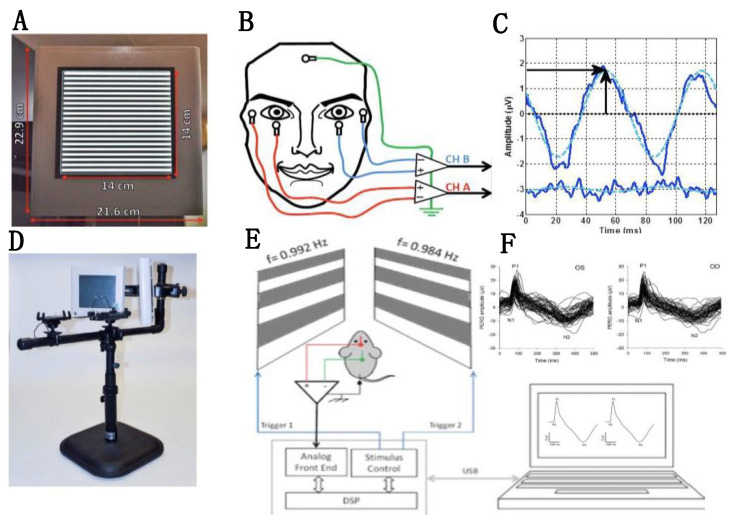
Recording set up for human and mouse PERG. (**A**,**D**) Patterned visual stimuli generated on LED tablets. (**B**) Surface electrode placement in human subjects for binocular PERG recording. (**C**) Example of human PERG and noise waveforms in response to fine gratings alternating 15.6 times/s. The blue dashed line superimposed to the PERG waveform represents the 15.6 Hz response component isolated in the frequency domain that is actually measured. Amplitude is measured from zero to the positive peak (vertical arrow), and latency is the corresponding time-to-peak (horizontal arrow). (**E**) Dual stimulus for binocular PERG recording in the mouse using subcutaneous needle electrodes in the snout and on the back of the head. (**F**) Example of mouse PERG waveforms response to large gratings alternating at slight different rates for each eye around 2 times/s. Amplitude is measured from the positive wave P1 to the negative wave N2, and latency is the time-to-peak of the positive wave P1. In the example, PERGs from different normal C57Bl/6J mice are superimposed to emphasize reproducibility. Figure panels are replotted from Monsalve et al., TVST 2017 and Chou et al., IOVS 2014.

**Figure 2 cells-10-01398-f002:**
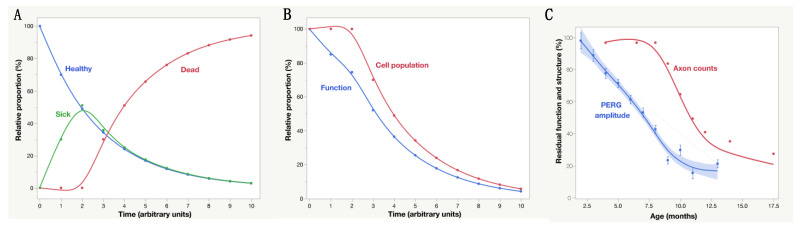
(**A**) Model of progressive optic neuropathy in which a given proportion (30%) of healthy RGCs become sick at time 1 and survive until time 2, with the process repeating at each successive time unit. (**B**) Assuming that sick RGCs function at 50% of normal capacity, the overall RGC function as measured by PERG decreases exponentially over time, anticipating a similar decay of the mean inner retinal thickness as measured by OCT. **C**: Natural history of PERG amplitude and optic nerve axon counts in DBA/2J mouse glaucoma. PERG data are replotted from Saleh et al., 2007, and represent the mean ± SEM of 32 mice; average axon counts are derived from Libby et al. 2005 and Anderson et al. 2005. The shaded area represents the 95% confidence interval of the spline regression curve interpolating PERG data.

**Figure 3 cells-10-01398-f003:**
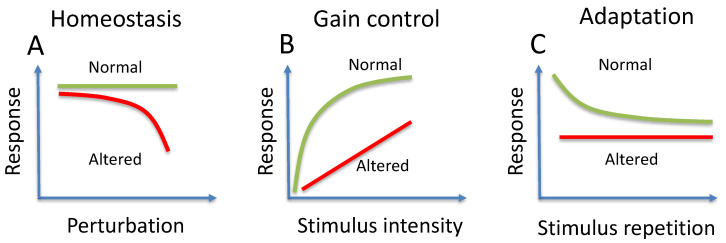
Conceptual examples of normal (green lines) and altered (red lines) homeostasis (**A**), gain control (**B**), and adaptation (**C**).

**Figure 4 cells-10-01398-f004:**
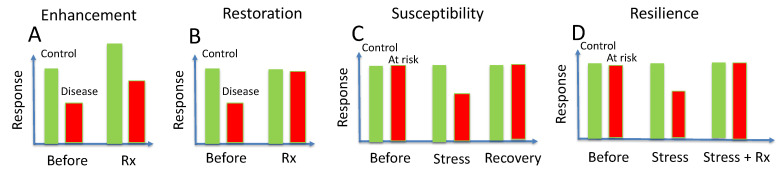
Conceptual examples of enhancement (**A**), restoration (**B**), susceptibility (**C**) and resilience (**D**) in control eyes (green bars) and diseased/at risk eyed (red bars).

**Figure 5 cells-10-01398-f005:**
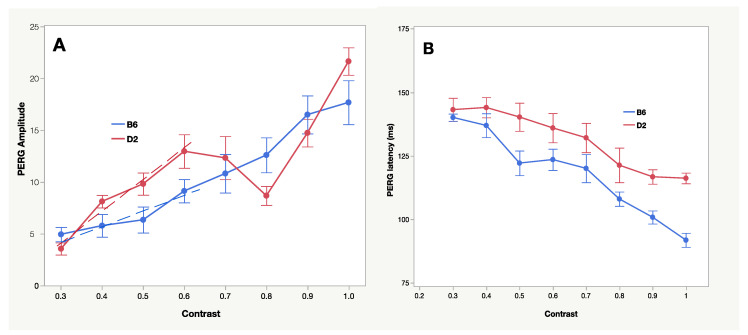
Contrast transfer function of PERG amplitude (**A**) and latency (**B**) in different mouse strains. B6: C57BL/6J; D2: DBA/2J. A: Different slopes of dashed lines represent different contrast gains of B6 and D2 PERGs. The dip in response amplitude at 0.8 contrast in the D2 contrast function indicates that generators with different spatio-temporal properties interact resulting in a reduction in the summed response. Replotted from Porciatti et al., 2010.

**Figure 6 cells-10-01398-f006:**
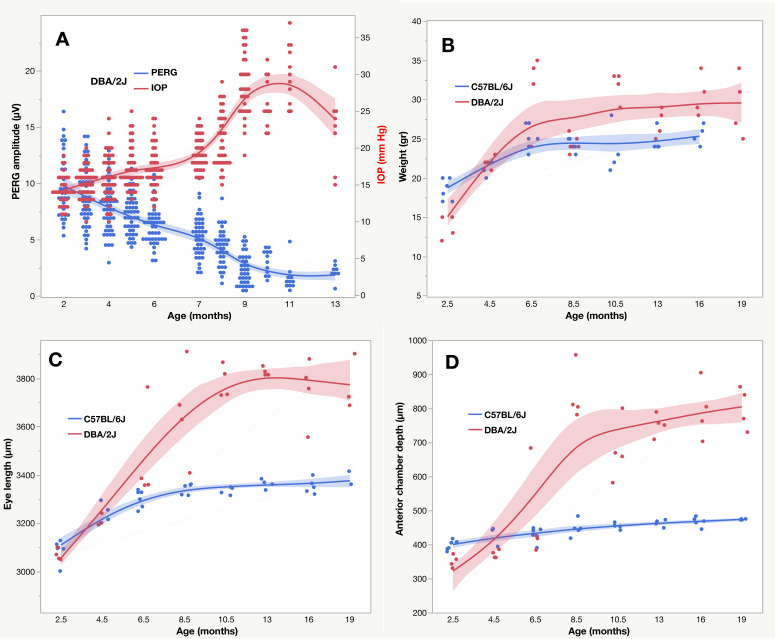
Age-related changes in C57BL/6J and DBA/2J mice. (**A**) PERG amplitude and IOP; (**B**) body weight; (**C**) Eye length; (**D**) anterior chamber depth. Panel (**A**) is replotted from Saleh et al., 2007 and panels (**B**–**D**) from Chou et al., 2010. The shaded areas represent the 95% confidence interval of the spline regression curves.

**Table 1 cells-10-01398-t001:** Conceptual stages of retinal ganglion cell (RGC) death in glaucoma and optic neuropathies that can be tested with non-invasive electrophysiology such as pattern electroretinogram (PERG) and imaging such as optical coherence tomography (OCT) and scanning laser ophthalmoscopy (SLO) combined with apoptotic markers.

**Hypothetical Transitional Stages of RGCs in Degenerative Optic Neuropathies**						
0	1	2	3	4	5	6
Healthy	At Risk	Sick	Very sick	Dying	Dead	Effaced
**Non-Invasive Electrophysiology**						
Normal	±Abnormal	Abnormal, Reversible	Abnormal, Reversible	Floor, Irreversible	Floor	Floor
**Non-Invasive Imaging**						
Normal	Normal	Normal	Normal	Abnormal	Abnormal	Abnormal

**Table 2 cells-10-01398-t002:** Relevant attributes of RGC function that can be investigated with PERG dynamics before and after neuroprotective treatment.

Neuronal Processes Altering PERG Dynamics	
Homeostasis	Stable function despite perturbance
Gain control	Reduction of sensitivity for high intensity stimuli
Adaptation	Reduction of response for repeated stimuli
Susceptibility	Temporary loss of function upon stress
Resilience	Reduced susceptibility to stress
Enhancement	Nonspecific improvement of function
Restoration	Recovery of lost function
Protection	Prevention of future loss of function
Rescue	Salvage of residual function
Plasticity	Reconfigured function

## Data Availability

Not applicable.

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
