# Peer review of "Modeling Retinal Ganglion Cell Dysfunction in Optic Neuropathies"

_cells, 2021, doi:10.3390/cells10061398_

Round 1

Reviewer 1 Report

This is an interesting paper. The authors try to offer a conceptual framework to identify progressive stages of retinal cell dysfunction in mouse models of glaucoma and other optic neuropathies. However, the present version of the manuscript does not seem to provide any model that can be quantified or tested.

General comments:

  • Title does not reflect the content of the manuscript.
  • Abstract should be rewritten and include more information about the proposed stages of cell death and the suggested electrophysiological methods to quantify RGC dysfunction.
  • The rationale for using the proposed 6 stages of cell damage are unclear. More information should be provided.
  • Statements such as “perhaps abnormal”, “moderately abnormal”, etc., should be avoided.
  • Equations of the proposed models should be included.
  • Table 2 should include information about how PERG dynamics can be used to investigate all the proposed RGC functions.
  • In general, more quantitative (and objective) information should be provided.
  • The discussion is missing. At least some advantages/disadvantages of the proposed model, from a clinical perspective, should be included.

Reviewer 2 Report

In this review article, the authors discuss the use of pattern electroretinography as a potential way to stage glaucoma according to a hypothetical model that defines early injury states for RGCs, during which neuronal dysfunction is present but cell death is not imminent and may be reversible.  They also discuss several theoretical responses that may be evident in subjects undergoing PERG depending on stressor or therapeutic exposures, and present empiric data from published studies that are exemplary of these responses.

Overall this is an interesting manuscript and describes the RGC death process in terms of early reversible stages of dysfunction that may be identifiable through electroretinography.  The ideas presented here are intuitive, and are laid out in a very clear manner with excellent examples from the literature for support.  I have a few suggestions and concerns regarding the manuscript as presently written:

On line 38-39, the authors state that “each stage is identifiable in vivo with non-invasive electrophysiology and/or non-invasive imaging” but then on lines 42-43 it is stated that RGC dysfunction at stages 1-3 is “potentially” identifiable with electrophysiology but it is unlikely that will be detected with imaging.  Please clarify – the sentence on line 38-39 seems overly certain of the benefit of electrophysiology in early stages of the disease if they have not yet been proven or are in widespread clinical application.

What is the evidence for some of the claims in table 1? Do we know that sick RGCs have reversible electrophysiological defects, or that stage 3 are potentially reversible? I think this aspect requires some further discussion with references.  Or if this is purely hypothetical, that may need to be stated more clearly.

The manuscript makes impressive claims about PERG, which seems to conflict with the fact that PERG is not in widespread clinical use by glaucoma clinicians, nor it is a routine part of outcomes for large clinical trials in glaucoma.  I think the reasons behind this deserve some attention here, and perhaps details regarding what research or improvements need to be done in order to make PERG more useful and widely adopted for managing human glaucoma patients. 

Line 79-80 – where in the scale from table 1 do Annexin-5 labeled apoptosing RGCs fit?  Is enough known to reference a specific stage?

I like Figure 1, where published data supports the model being introduced. Of note, line 103 talks about a shaded area to represent the 95% confidence interval of the regression, but this appears to have been omitted from the actual figure.  The same is true of figure 5.

I think this manuscript would benefit from further discussion that includes specific recommendations for future studies that could prove or disprove this model shown in Table 1 (in addition to referencing the existing data that is consistent with the model, which is very well done).

It is interesting that most of the measures discussed here, including PERG, provide a global outcome metric (i.e. amplitude), even though glaucoma tends to be a local disease (at least in early and mid stages).  The authors may wish to discuss differences between global measures versus local measures versus single cell measures, and where PERG fits. 

It might help an unfamiliar reader to show in a figure an example of the stimulus used for PERG, and an example of a PERG tracing.

Line 193 – to meet the definition of restoration, Lu et al would have also needed to show that OCT4, SOX2, and KLF4 does not alter the PERG of normal animals. Was that the case? If not shown, this could have been enhancement.

Line 216 – redundancy of the phrase “such as head-down posture” in this sentence.

Author Response

Please see the attachment. As some answers were relevant to both Reviewers, response to Reviewers were merged in one file

Round 2

Reviewer 2 Report

The authors have adequately addressed my prior questions and concerns, and I feel the quality of the manuscript is improved.